# Exploring the Association between Low-Density Lipoprotein Subfractions and Major Adverse Cardiovascular Outcomes—A Comprehensive Review

**DOI:** 10.3390/ijms24076669

**Published:** 2023-04-03

**Authors:** Laura Adina Stanciulescu, Alexandru Scafa-Udriste, Maria Dorobantu

**Affiliations:** 1Faculty of Medicine, “Carol Davila” University of Medicine and Pharmacy, 050474 Bucharest, Romania; 2Cardiology Department, Emergency Clinical Hospital Bucharest, 014461 Bucharest, Romania; 3Romanian Academy, 010071 Bucharest, Romania

**Keywords:** lipoproteins, LDL, subfractions, cardiovascular risk, dyslipidemia, coronary artery disease, atherosclerosis

## Abstract

Cardiovascular disease (CVD) impacts hundreds of millions of people each year and is the main cause of death worldwide, with atherosclerosis being its most frequent form of manifestation. Low-density lipoproteins (LDL) have already been established as a significant cardiovascular risk factor, but more recent studies have shown that small, dense LDLs are the ones more frequently associated with a higher overall risk for developing atherosclerotic cardiovascular disease. Ever since atherogenic phenotypes were defined for the first time, LDL subfractions have been continuously analyzed in order to identify those with a higher atherogenic profile that could further become not only high-accuracy, effective prognostic biomarkers, but also treatment targets for novel lipid-lowering molecules. This review sets out to comprehensively evaluate the association between various LDL-subfractions and the risk of further developing major adverse cardiovascular events, by assessing both genetical and clinical features and focusing on their physiopathological characteristics, chemical composition, and global ability to predict long-term cardiovascular risk within the general population. Further research is required in order to establish the most beneficial range of LDL-C levels for both primary and secondary prevention, as well as to implement LDL subfraction testing as a routine protocol, separately from the general assessment of the other traditional cardiovascular risk factors.

## 1. Introduction

Atherosclerosis is the most frequent form of cardiovascular disease (CVD), consisting mainly of lipid buildup and inflammatory processes within the walls of large arterial vessels, which ultimately leads to major cardiovascular events such as acute coronary syndromes (ACS) and stroke. In addition to being the leading cause of mortality worldwide, atherosclerosis also places an enormous burden on healthcare systems across the globe. Thus, there is an increasing need to identify high-risk individuals at an early stage, before they become clinically symptomatic.

Low-density lipoproteins have been identified as a major prognostic factor in the development and further evolution of atherosclerosis and cardiovascular disease. There are several subclasses of LDL-C, including large floating (lb), intermediate, and small, dense (sd) LDLs [1]. Recent studies have shown that sdLDL is more atherogenic than other LDL subfractions and that sdLDL-C is a higher-accuracy prognostic biomarker for overall CVD than total LDL-C. There are currently many laboratory methods available for separating LDL subclasses, although the results obtained by these methods are extremely variable at the moment. Although the latest research shows promising results, further studies are needed to establish a series of standardized methods and guidelines in order to evaluate sdLDL subfractions and properly adjust the current clinical practice.

This study sets out to systematically evaluate the available evidence concerning the associations between small, dense low-density lipoprotein (sdLDL) and major adverse cardiovascular events (MACE), taking into consideration both genetic and clinical aspects and focusing on LDL-C physiopathological characteristics, chemical composition, and overall capability of predicting long-term cardiovascular risk in the general population.

The literature search was conducted in various databases, including PubMed, MEDLINE, and the Cochrane Library. The focus was on English-language sources and studies in human subjects, but in vitro, experimental studies were also taken into consideration. Additional information was requested from the authors when it was considered necessary.

## 2. Role of Low-Density Lipoproteins in Cardiovascular Disease

Lipoproteins in plasma are responsible for the transportation of lipids to tissues, which can be used for energy production, the storage of lipids, the production of steroid hormones, and the formation of bile acids. Lipoproteins consist of cholesterol, triglycerides, phospholipids, and proteins (apolipoproteins), which serve as structural elements, ligands for cellular receptors, and activators/inhibitors of enzymes. The six main lipoproteins in the blood are chylomicrons, very low-density lipoprotein (VLDL), intermediate-density lipoprotein (IDL), LDL, lipoprotein a (Lp(a)), and HDL [1]. The physical and chemical characteristics of human plasma lipoproteins are presented in Table 1.

Apolipoprotein B (ApoB), containing lipoproteins of less than 70 nanometers in diameter, can traverse the endothelial barrier, particularly when the endothelium is compromised, where they may become ensnared following interactions with extracellular components such as proteoglycans. Subsequently, they are retained within the arterial wall and further begin a very complex process that ultimately leads to an atheromatous plaque.

Those with higher levels of ApoB-containing lipoproteins in their plasma will take in more particles, build up lipids more quickly, and grow more quickly, leading to the advancement of atherosclerotic plaques. Therefore, the overall atherosclerotic burden is likely to be at least partly defined by the serum-circulating levels of LDL-C and other ApoB-based lipoproteins and also by the period of exposure to these particles.

The development of atherosclerotic plaques starts with damage to the endothelial cells, allowing more LDL particles to pass through the walls of the vessels. These lipoproteins, particularly LDL, become caught in the intima by the cellular matrix. The LDL is then modified and taken in by macrophages using special cell-surface pattern recognition receptors (also known as scavenger receptors), leading to the creation of foam cells. As more lipids accumulate, smooth muscle cells move into the lesion, encapsulating the plaque, and forming a fibrous barrier to protect the lipid core from the lumen of the vessel. In some cases, these plaques can lead to a decrease in blood flow, or the plaque can rupture, causing a thrombus to form and block the flow of blood.

The entire process of LDL oxidation is schematically described in Figure 1.

The overall atherogenic particularities of small, dense LDL particles are depicted in Table 2, as a series of sequential processes that increase the overall atherogenic potential of these molecules [2].

As the atherosclerotic plaque burden increases and its composition changes, it reaches a tipping point where the disruption of the plaque can occur, forming an overlying thrombus that blocks the flow of blood, which can further lead to either an acute coronary syndrome or even sudden death. As more ApoB-containing lipoproteins are retained, the risk of having an acute atherosclerotic cardiovascular disease (ASCVD) event rapidly increases. Maintaining a healthy lifestyle is critical to reducing ApoB-containing lipoprotein levels, thereby slowing the progression of atherosclerosis and helping to prevent major cardiovascular events. It is important to recommend treatments that can lower LDL-C and other ApoB-containing lipoproteins for both primary and secondary prevention.

## 3. What We Know So Far

The latest ESC/EAS Good Clinical Practice Guidelines for the Management of Dyslipidaemias (2019) recommend an LDL-C reduction of at least 50% from baseline and an LDL-C goal of less than 55 mg/dL (1.4 mmol/L) for both primary and secondary prevention in very high-risk individuals, as well as for primary prevention in very high-risk individuals with familial hypercholesterolemia. The guidelines also recommend LDL-C goals of less than 40 mg/dL (1 mmol/L) in patients with ASCVD who experienced a second vascular event within 2 years while on a maximally tolerated statin treatment, less than 70 mg/dL (1.8 mmol/L) for overall prevention in high-risk individuals, less than 100 mg/dL (2.6 mmol/L) in moderate risk individuals, and less than 116 mg/dL (3 mmol/L) in those at low risk of developing a major cardiovascular event.

The lipid goals and lipid-lowering therapeutic strategy are part of a larger, more comprehensive cardiovascular (CV) risk reduction management approach and it was previously believed that it was appropriate to reduce the LDL-C serum levels to as low as possible in high- and very high-risk patients.

We have multiple pieces of evidence to show that lowering the LDL-C beyond the recommended goals can further reduce the number of ASCVD events [3,4,5]. One of the first big trials to confirm this theory was the Treating to New Targets (TNT) study, which compared atorvastatin 10 mg and 80 mg daily in a double-blind fashion in 10,001 patients with previously objectified coronary artery disease (CAD). The mean LDL-C levels during the study were 101 mg/dL (2.6 mmol/L) and 77 mg/dL (2.0 mmol/L) in the 10 mg and 80 mg groups, respectively, which led to a 22% relative reduction in the primary composite endpoint of CAD (*p* < 0.001) over a total follow-up period of 6 years [6].

The Study of the Effectiveness of Additional Reductions in Cholesterol and Homocysteine (SEARCH) study compared the effects of two different doses of simvastatin (20 mg and 80 mg) on 12,064 people who had previously suffered a heart attack in a double-blind trial. After two months, the 80 mg group had a 0.51 mmol/L (19.7 mg/dL) lower LDL cholesterol level than the 20 mg group, but this difference was reduced to 0.29 mmol/L (11.2 mg/dL) after five years. There was a nominal reduction in nonfatal myocardial infarctions in the 80 mg group, but the primary endpoint of major vascular events was reduced by only 6% (*p* = 0.10) [7].

A meta-analysis published later based on similar studies revealed a 15% reduction in risk with a mean decrease in LDL cholesterol of 0.51 mmol/L (19.7 mg/dL) when using high-dose statins (*p* < 0.0001) [3].

On the other hand, more recent studies propose a different hypothesis. A study published in 2020 by C.D.L. Johannesen et al. prospectively evaluated 108,243 subjects with a median follow-up period of 9.4 years in order to evaluate the correlation between the serum levels of LDL-C and all-cause mortality, and concluded that the association between LDL-C and the risk for all-cause mortality was U-shaped, with both low and high levels associated with an increased risk of mortality (the lowest overall risk being observed at an LDL-C concentration of about 140 mg/dL–3.6 mmol/L) [8].

Another prospective study based on a cohort of 14,035 adults aged 18 years and older, with a median follow-up period of 23.2 years (with a mean age of 41.5 years, 51.9% women), noted that both very low and very high LDL-C levels were associated with increased risk of CVD mortality. In particular, very low LDL-C levels were associated with a higher risk of stroke and all-cause mortality [9].

Another study evaluated the association between low levels of LDL-C and intracerebral hemorrhage (ICH) and concluded that, in patients who present a high risk of developing ICH, a cautious approach and individualized therapy strategy are advised before considering an aggressive LDL-C level reduction approach, as the association still remains uncertain, even after a comprehensive literature review. This particularly relates to the new cholesterol-lowering treatments that have emerged lately, and those that are currently in development, which are created to achieve swift and effective lowering of LDL-C with improved clinical outcomes and less excessive ICH adverse events [10].

Furthermore, a recent study by Yen et al. (2022) [11] focusing on assessing the benefit of LDL-C level lowering treatment and the overall cardiovascular and renal outcomes in patients with stage 3 chronic kidney disease (CKD) evaluated approximately 8500 newly diagnosed CKD patients and divided them into three groups according to their first LDL-C level after the index date: <70 mg/dL, 70 to 100 mg/dL, and >100 mg/dL. Compared with the LDL-C ≥ 100 mg/dL group, the 70 ≤ LDL-C < 100 mg/dL group exhibited significantly lower risks of major adverse cardiac and cerebrovascular events (6.8% versus 8.8%; subdistribution hazard ratio (SHR) 0.76 (95% CI, 0.64–0.91)), intracerebral hemorrhage (0.23% versus 0.51%; SHR 0.44 (95% CI 0.25–0.77)), and new-onset end-stage renal disease requiring chronic dialysis (7.6% versus 9.1%; SHR 0.82 (95% CI 0.73–0.91)). On the other hand, the LDL-C < 70 mg/dL category exhibited a slightly lower risk of major adverse cardiac and cerebrovascular events (7.3% versus 8.8%; SHR 0.82 (95% CI 0.65–1.02)) and a remarkably lower risk of new-onset end-stage renal disease requiring chronic dialysis (7.1% versus 9.1%; SHR 0.76 (95% CI 0.67–0.85)). Although considerable progress is being made at the moment regarding novel lipid-lowering agents and therapeutic strategies, it is still uncertain whether an aggressive targeted approach to lipid management still remains the optimal choice for all patients, regardless of their overall CV risk, and the discussion becomes all the more nuanced as the patients associate other comorbidities.

## 4. Genetics

Genetic influences on serum lipoprotein levels have been studied for many years now, and a reference study from as early as 1990 demonstrated that the proposed genetic locus accountable for LDL subfraction phenotypes eventually results in an atherogenic lipoprotein phenotype. Thus, two distinct phenotypes (namely, A and B) were presented, which were identified through the non-denaturing gradient gel electrophoretic analysis of LDL subclasses. While phenotype A was defined by large, floating LDL particles, phenotype B was characterized by a predominance of small, dense particles, and the latter was further associated with a higher risk of myocardial infarction [12].

Recent advances in technology have enabled the identification of a variety of genetic variants, from extremely rare to more common, that have substantial impacts, from major to minor, on the dissimilarities in the plasma lipid and lipoprotein levels among individuals.

Monogenic dyslipidemias are mainly defined by the primary lipid or lipoprotein disrupture: higher or lower concentrations of LDL-C or HDL-C, or elevated triglycerides (TG) [13]. At the moment, there are 27 clearly identified monogenic dyslipidemias, characterized by unusually high or low levels of plasma lipids or lipoproteins, with distinct clinical manifestations due to numerous genetic mutations impacting a total of 25 genes [14].

The monogenic dyslipidemias and dyslipoproteinemias that we are currently aware of, including the genes that encode them and the chromosomes involved in the disruptive mechanisms, are described in Table 3.

Familial hypercholesterolemia (FH), probably the most common monogenic dyslipidemia, is regulated by three primary genes (LDL-R, APOB, and PCSK9) and can cause premature atherosclerosis and cardiovascular complications. However, in some individuals, the FH symptoms are linked to variants of other genes. It was previously believed that it occurs in about 1 in 500 people, but more recent studies have shown that the real prevalence of 1 in 127 individuals is actually established by the LDL receptor gene pathogenic variant carrier status. There are currently nine genes that underlie FH and FH-like phenotypes; in addition to the before-mentioned well-known causative genes for co-dominant forms of FH and LDLRAP1 or ARH involved in the purely recessive FH, we should also mention the APOE p.LEU167del variant (which causes a dominant form of FH), ABCG5, ABCG8, and LIPA, which ultimately lead to an FH-like phenotype [15].

New potential FH loci have recently been studied using exome sequencing, with promising results. PNPLA5 carriers, especially those with a rare or low-frequency variant, have a higher risk of developing extreme serum levels of LDL-C, although vertical transmission in families or mechanistic impairment was not demonstrated [16]. Other newly identified genes with a potential implication in the disrupting mechanisms involved in FH phenotypes include CH25H and INSIG2, but further studies are needed in order to validate them [17].

More recent studies have clarified the role of polygenic determinants in FH and even proposed a screening score to distinguish between patients with FH from healthy subjects, concluding that in 88% of mutation-free patients, the hypercholesterolemia is most likely to have a polygenic basis [18]. About 40% of clinically diagnosed individuals with heterozygous forms of FH (HeFH) do not have a monogenic mutation, but rather have gained an atherogenic burden of LDL-C-raising single nucleotide polymorphism (SNP) alleles that ultimately raise their LDL-C levels up to the HeFH range. This effect could explain the high FH-like LDL levels in patients without any clearly identified monogenic mutation, despite adequate genetic testing, which is exactly why screening for polygenic factors should be included in all FH molecular cases [19,20].

## 5. Importance of Dosing LDL Subfractions and How It Impacts Overall Cardiovascular Risk

While it may be true that a complete, standardized lipid panel should be performed for a first-time estimation of the overall ASCVD risk, dosing the plasmatic levels of ApoB and LDL subfractions could allow for a more complex evaluation of the atherogenic burden and could also provide a proper starting point for therapeutic management. Recent studies have shown that dosing the LDL particle subclasses has the potential for the early detection of certain atherogenic lipoprotein patterns that are below the discrimination level of standardized testing and can therefore identify those at a higher risk of major cardiovascular events before they become clinically symptomatic [21,22,23,24,25].

Multiple types of research evaluated the significance of dosing LDL subfractions in estimating the risk of further developing major cardiovascular events. The Quebec Cardiovascular Study showed that small LDL subfraction levels were independently correlated with coronary heart disease (CHD) risk in 2072 men over a 13-year follow-up period. Contrariwise, large LDL particles were proven to have no predictive value in this matter [26]. Two other prospective studies, namely, the Atherosclerosis Risk in Communities (ARIC) and the Multi-Ethnic Study of Atherosclerosis (MESA), proved a directly proportional relationship between small, dense LDL-C levels and the risk for ischemic heart disease. As previously shown by the Quebec Cardiovascular Study, no relationship with large LDL particles was found [27,28]. The Stanford Five Cities Project [29] and the Physician’s Health Study [30] also proved that a small LDL-C diameter is an important univariate predictor for coronary artery disease (CAD). The same findings were also confirmed by many other subsequent research projects and ultimately reviewed by the European Panel of Experts while discussing the pathophysiology, atherogenic potential, and clinical importance of LDL-C subfractions [31,32].

Given their size and diameter, small, dense LDL (sdLDL) particles have a higher chance of entering arterial tissue than larger subspecies, indicating greater permeability through the endothelium. Additionally, these particles have a significantly lower receptor-mediated uptake, a higher binding to proteoglycans, and are more prone to oxidation, which eventually leads to a modified surface lipid layer caused by decreased free cholesterol, fewer antioxidants, and higher amounts of polyunsaturated fatty acids [33,34].

SdLDL-C subfractions are highly susceptible to oxidation, and this could be due to their configuration, as they carry fewer antioxidative vitamins. The plasmatic oxidative process ultimately leads to various specific epitopes on the surface of LDL particles that generate an increased immune and inflammatory response.

There is a known correlation between elevated levels of lipoprotein-associated phospholipase A2 (Lp-PLA2) in LDL particles and cardiovascular disease. Higher PLA2 levels have been found in electronegative LDL as well as in more progressed atherosclerotic plaques. Inside the lipoprotein particle, this enzyme cuts apart oxidized phospholipids, liberating proinflammatory substances and further raising its atherogenic potential [35,36].

An additional atherogenic alteration of LDL particles is desialylation, a process performed by trans-sialidase, a glycosylphosphatidylinositol (GPI)-anchored surface enzyme that plays an important part in the metabolism of glycoconjugates. Long-term contact with blood plasma leads to a progressive desialylation of the particles, which already have a decreased sialic acid content in comparison to large buoyant LDL (lbLDL) in subjects with an atherogenic phenotype B. As the desialylation process creates a higher affinity between the sdLDL particles and proteoglycans, the final products (desialylated sdLDL molecules) can resist within the subendothelial space for a longer period, thus leading to an increased lipid storage and atherosclerosis plaque development [37,38,39].

The various characteristics of sdLDL particles make them highly atherogenic and related to the beginning stages of subclinical atherosclerosis and endothelial dysfunction, which increases the risk of cardiovascular events. Research has revealed that the predominance of sdLDL particles is associated with a higher risk of CAD, as shown in large epidemiological and clinical intervention trials. Consequently, sdLDL is now considered a novel, high-accuracy biomarker for assessing the overall ASCVD risk, as it is a significant lipid abnormality observed in individuals with CAD, peripheral arterial disease (PAD), diabetes, metabolic syndrome, and other cohorts with a global high cardiovascular risk [40,41,42,43,44].

## 6. Future Directions

As we are dealing with a cardiovascular disease epidemic, the need to identify and develop novel prognostic biomarkers that could predict the overall cardiovascular risk with higher accuracy has recently become a public health issue and a high-interest area of research. While it is certain that we currently possess various prognostic markers in order to establish each patient’s CVD risk, we are still facing the need to perform a more effective screening within the general population in order to identify high-risk individuals who might benefit from primary prevention programs more quickly. That would not only significantly decrease their chance of further developing a major cardiovascular event, but would also alleviate the burden that is currently placed on public health systems worldwide.

Even though we are well aware of the importance of traditional cardiovascular risk factors, we are nevertheless struggling to understand why there are different outcomes in individuals with the same risk profile, according to the classic risk frames.

Whilst the genetic component is clearly of considerable substance and should definitely be taken into consideration, researchers have studied the different atherogenic profiles of LDL-C subfractions alongside other biomarkers that were proven to be involved in the early atherosclerotic process. Multiple studies (such as those we have previously mentioned) on various populational groups with different characteristics have shown that sdLDL possesses a series of particularities that make it more susceptible to oxidation and therefore more prone to activate a more nuanced inflammatory response within the arterial vessels. SdLDL was also associated with a higher chance of developing a major cardiovascular event, especially acute coronary syndromes, and has been linked to a higher global cardiovascular risk.

A better understanding of LDL-C subfractions’ different atherogenic potentials could not only refine the current cardiovascular risk scores to better predict each patient’s long-term risk, but also allows for a more tailored approach to establishing the proper prophylactic and therapeutic management for each individual.

Some new classes of oral antidiabetic drugs, such as GLP1-Ras, were shown to lower sdLDL levels with favorable outcomes in the long term, although their mechanisms are only relatively understood at the moment. On the other hand, cholesterol ester transfer protein inhibitors (CETPi) showed contrasting results—evacetrapib, either alone or in association with statins, exhibited an important lowering of ApoB-derived particles (including sdLDL), while anacetrapib caused a paradoxical increase in sdLDL while decreasing the number of total LDL particles and all of the other LDL subfractions [45,46,47].

As new molecules are currently under development (such as inclisiran, the novel PCSK9 inhibitors, and the small interfering RNA molecules that are being studied at the moment), it remains clear that further progress should be made in order to simplify and raise the effectiveness of future therapeutic options and ultimately translate them into quantifiable clinical benefits.

Gene-editing lipid-lowering technologies are also being studied at the moment, with impressive preliminary results. In a recently published study, CRISPR base editors that were delivered in vivo were shown to effectively modify disease-related genes in living cynomolgus monkeys, with significant reductions in the PCSK9 and LDL-C serum levels after a single-dose treatment [48]. As spectacular as it seems, this is definitely just the beginning in what seems to be the start of implementing a series of revolutionary medium- and long-term lipid-lowering therapies on a large scale.

Randomized controlled trials have already proven that lowering LDL-C levels can reduce the overall risk of CVD, with an established gradient between the degree of LDL-C decrease and the extent of CVD prevention [49]. Therefore, it is understandable that modern research should be more focused on intensifying the available screening methods to promptly identify those who might benefit the most from comprehensive primary prevention programs.

Further studies are needed to establish whether sdLDL can be considered a proper prognostic biomarker on its own and evaluate its potential for refining the cardiovascular risk algorithms and prediction charts that are currently in use.

## 7. Current Limitations

The evaluation of LDL subfractions, although a research area of interest at the moment, is not yet practiced regularly, mainly because it requires extensive infrastructure and leads to high processing costs. Even though many studies have shown a clear correlation between sdLDL and the risk of developing major cardiovascular events in the long term, independently from other traditional risk factors, further research is required to establish whether we can consider sdLDL a novel cardiovascular prognostic biomarker per se.

A greater number of LDL particles has been correlated with a higher chance of developing cardiovascular disease, but there are still various perspectives regarding the potential incremental benefit that measuring sdLDL levels would add to traditional risk factor assessment [50].

Despite considerable evidence that sdLDL-C concentration is a significant biomarker for predicting overall cardiovascular risk, there are still many questions that remain to be answered, and we believe it is too early to routinely utilize the currently available LDL subfraction tests to predict the risk of cardiovascular disease on a regular basis without taking into consideration other already validated prognostic markers.

Additional research on multiple, varied cohorts with different risk factors and comorbidities is needed to clarify and establish once and for all the impact of each LDL subfraction in the development of atherosclerosis and the global prediction of cardiovascular disease [51].

## 8. Conclusions

LDL-C and especially sdLDL particles have recently become one of the most viable predictors of ASCVD, as they offer an additional cardiovascular risk stratification value. Higher levels of sdLDL appear to correlate positively with an increased overall risk for developing a major cardiovascular event and are a common finding in individuals with type 2 diabetes mellitus, metabolic syndrome, CAD, and PAD. These facts show how crucial both the quantity and quality of LDL are for properly managing cardiometabolic risk.

The results of recent studies demonstrate that LDL fractions have different atherogenicity, with sdLDL being more atherogenic than larger LDL subfractions. Multiple research projects set out to evaluate the role of sdLDL particles in the development of ASCVD, but we are currently facing notable variations between different methods. The development of an inexpensive, fast, and dependable method of quantitative LDL subfraction analysis is very much needed.

More research is required to establish the most beneficial range of LDL-C levels for both primary and secondary prevention within the general population, as well as to implement LDL subfraction testing as a routine protocol, separately from the general assessment of the other traditional cardiovascular risk factors.

This area of interest is currently in continuous development and allows for multiple avenues of further research in order to elucidate the physiopathological mechanisms, improve therapeutic approaches, and bring about tangible results in the clinical setting.

## Figures and Tables

**Figure 1 ijms-24-06669-f001:**
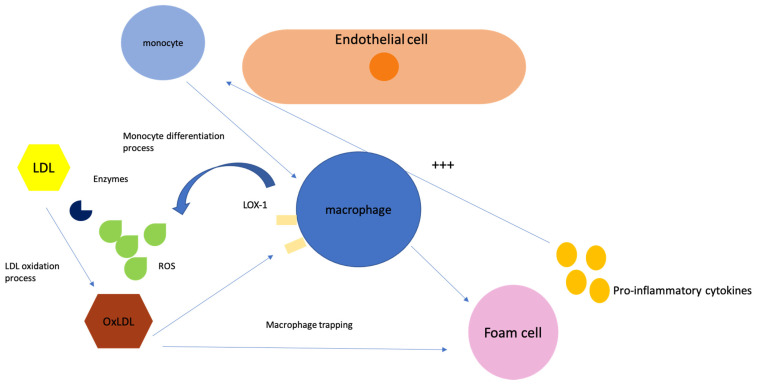
LDL-C oxidation process—within the sub-endothelial extracellular matrix, monocytes differentiate into macrophages that further express a series of scavenger receptors (including LOX-1 as depicted in the before-mentioned figure, alongside CD36 and toll-like receptors). Scavenger receptors generally have a high affinity for oxidized LDL (OxLDL) and are not downregulated, which ultimately leads to a significant intracellular lipid buildup. The figure also shows the macrophage trapping process, consisting of a vicious circle based on the differentiation into macrophages that promotes inflammatory processes, involving cell retention, accelerated LDL oxidation, and the recruitment of more macrophages. This eventually leads to foam cell formation and, eventually, the aggregation of OxLDL promotes foam cell apoptosis or even necrosis, creating cellular debris that is deposited within the atherosclerotic plaque and contributes to inflammatory advancement. Abbreviations: LDL, low-density lipoprotein; OxLDL, oxidized low-density lipoprotein; LOX-1, lectin-like oxidized low-density lipoprotein receptor-1; ROS, reactive oxygen species.

**Table 1 ijms-24-06669-t001:** Physical and chemical characteristics of human plasma lipoproteins.

Plasmatic Lipoprotein	Density (g/mL)	Source	Composition (%)		Apolipoproteins	
			Lipid	Protein	Major	Others
Chylomicrons	<0.95	intestine	98–99	1–2	ApoB-48	ApoA-I, A-II, A-IV, A-V
VLDL	0.95–1.006	liver	90–93	6–8	ApoB-100	ApoA-I, C-II, C-III, E, A-V
IDL	1.006–1.019	catabolism of VLDL	89	11	ApoB-100	ApoC-II, C-III, E
LDL	1.019–1.063	catabolism of VLDL via IDL	79	21	ApoB-100	-
HDL	1.063–1.210	liver, intestine, catabolism of CM and VLDL	67	33	ApoA-I	ApoA-II, C-III, E, M
Lp(a)	1.006–1.125	liver	80	20	Apo(a)	ApoB-100

The physical and chemical overall characteristics of human plasma lipoproteins, adapted from the 2019 ESC/EAS Guidelines for the Management of Dyslipidaemias [1].

**Table 2 ijms-24-06669-t002:** Overall characteristics of small, dense low-density lipoproteins (sdLDL) summarized as a series of sequential processes.

sdLDL Properties	Sequence of Processes
Reduced binding to the LDL-C receptor	Increased residence time
Increased penetrance of the arterial wall	Increased infiltration
Increased affinity for arterial proteoglycans	Increased sequestration
Increased susceptibility to oxidation	Increased oxidation
Increased total cholesterol deposits	Accelerated atherosclerosis process

**Table 3 ijms-24-06669-t003:** Monogenic dyslipidemias and dyslipoproteinemias.

Phenotype	Disorder	Genes Involved	Chromosome	References
High LDL-C	Familial hypercholesterolemia	*LDLR*	19p13.3	[14,15]
	Familial defective apolipoprotein B	*APOB*	2p24-p23	[14,15]
	Autosomal dominant hypercholesterolemia type 3 (*PCSK9* gain of function)	*PCSK9*	1p32.3	[14,15]
	Autosomal dominant hypercholesterolemia type 4	*STAP1*	4q13.2	[14,15]
	Autosomal dominant hypercholesterolemia type 5	*APOE*	19q13	[14,15]
	Autosomal recessive hypercholesterolemia	*LDLRAP1 (ARH)*	1p36-p35	[14,15]
	Cholesterol ester storage disease	*LIPA*	10q21.31	[14,15]
	Sitosterolemia	*ABCG5/ABCG8*	2p21	[14,15]
Low LDL-C	Abetalipoproteinemia (Bassen–Kornzweig syndrome)	*MTTP*	4q24	[13,14]
	Hypobetalipoproteinemia	*APOB*	2p24-p23	[13,14]
	*PCSK9* deficiency with low LDL-C levels (*PCSK9* loss of function)	*PCSK9*	1p32.3	[13,14]
	Familial combined hypolipidemia (*ANGPTL3* deficiency)	*ANGPTL3*	1p31.1-p22.3	[13,14]
	Chylomicron retention disease (Anderson disease)	*SAR1B*	5p31.1	[13,14]

## Data Availability

Not applicable.

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
