# Peer review of "Exploring the Association between Low-Density Lipoprotein Subfractions and Major Adverse Cardiovascular Outcomes—A Comprehensive Review"

_ijms, 2023, doi:10.3390/ijms24076669_

Round 1
Reviewer 1 Report
Dear authors,
I have studied with great interest the manuscript “Exploring the Association between Low-Density Lipoprotein Subfractions and Major Adverse Cardiovascular Outcomes - A Comprehensive Review”.
The authors summarized in their review the information about the association between various LDL-subfractions and the risk of further developing major adverse cardiovascular events. A great pool of information has been analyzed and the authors gave the comprehensive understanding of those mechanisms. That will be definitely useful and interesting for readers.
The main question is addressed by the research was the physiopathological characteristics, chemical composition of various LDL-subfractions and their global ability to predict long-term cardiovascular risk within the general population.
The manuscript is clearly exposed and well written, the text clear and easy to read. The topic is original.
But I have some comments to improve the quality of the presentation.
1. For a better understanding of the review, a figure depicting the mechanisms of atherosclerotic plague formation could be provided.
2. Prove a summary table of dosing LDL-Subfractions and their impacts on the overall cardiovascular risk.
3. Check-out the paper for some misprints.
I express my gratitude to the authors for their great work done.
Author Response
Dear Reviewer,
Thank you for your kind and useful suggestions. As you can see in the newly uploaded revised manuscript, we have taken them into consideration and made the recommended rectifications. We have added a diagram describing the mechanisms involved in the atherosclerotic plaque formation, as well as a summary table regarding LDL subfraction dosing and their overall impact on the general cardiovascular risk. We have also checked the entire paper for misprints and corrected what was necessary.
Best regards,
Reviewer 2 Report
The manuscript (review) is about a health issue in relation to cardiovascular events related to cholesterol. In general, it is written in a simple, understandable way and identifies an area of interest: sd-LCD particles as an additional predictor of cardiovascular risk.
To improve the manuscript the following is suggested:
1) Include a photo or diagram of the formation process and the final appearance of the atherosclerotic barrier after paragraph 75-83 in which this process is described.
2) Include a photo, figure or diagram of sdLDL in which the configuration and most important dimensional characteristics are appreciated or specified, after paragraph 65-69 or in the area of the manuscript in which the authors consider most convenient.
3) Add bibliographical citations of the following paragraphs:
65-69; 70-74: 75-83; 94-103; 251-255; 256-264; 287-295
4) Line 64, after LDL change ";" by ",". Between brackets indicate what Lp(a) means
5) Choose only one way of citing, not both: if the author is mentioned at the beginning, include the numerical reference [12] at the end of the citation, or if the numerical reference is included at the end, remove the author's name from the beginning appointment. This erroneous way of double citing is found in paragraphs 127-133; 139-146; 147-160; 166-174.
6) Line 179. Add the meaning of TG
7) Table 1. Include in the table a fifth column with the reference of each one of the dyslipidemias
8) Line 257. Add the meaning of GPI
9) Line 260. Add the meaning of lbLDL
10) Line 274, Line 319, Line 341. Remove “:” at the end of the sentence
11) Write the text in the third person, not in the first person as in Line 275, Line 284, Line 351
12) Line 290. In the sentence “….Multiple studies…” include in parentheses after this sentence some of those studies.
13) Line 307 mentions “….As new molecules are currently under development…”, specify which molecules?; what are they being developed for?
Author Response
Dear Reviewer,
Thank you for noting these shortcomings and for the useful recommendations. We have made the suggested corrections, as follows:
- we have added a diagram describing the formation process of the atherosclerotic plaque as it is described in the beforementioned paragraph;
- we have included a diagram that points out the configuration and the most important dimensional characteristics of sdLDL subfractions;
- we added some bibliographical citations as you have recommended;
- we detailed on the full meaning of TG, GPI, lbLDL, Lp(a);
- we have corrected the situations were certain articled were double-cited;
- regarding the table describing the different types of familial dyslipidemias - we have added a separate column mentioning references for each type;
- regarding the writing in the first person - we made the suggested corrections, with the sole exception of the following sentence: ” Multiple research projects set out to evaluate the role of sdLDL particles in the development of ASCVD, but we are currently facing notable variations between different methods”, as we believe that writing in the first person in this specific situation better describes the fact that dealing with such variations in this field of research currently represents a challenge for all those interested in the subject;
- we have detailed on the multiple studies that are mentioned in line 290;
- we have detailed on the new molecules that are currently being under development that we have mentioned in line 307.
Best regards,
Reviewer 3 Report
The article is well written and contains important information about the association between low-density lipoprotein subfractions and adverse cardiovascular outcomes. I have a few remarks to make:
1) I missed the bibliographic citations in the introduction of the manuscript (where no source is mentioned) and during the writing of the work. The authors make various statements saying, including about several published works. What are the sources of these claims? In the case of a review, the bibliographic sources need to be better shown. Where are quotes numbered 35 to 39?
2) The authors should have given more value to the previous results published and compiled in the review, however, they emphasized more the limitations and stated that there are still few studies showing the association and use of prognostic tools.
Author Response
Dear Reviewer,
Thank you very much for your kind and extremely useful recommendations. Regarding the quotes numbered 35 to 39 - we have realized this was a gap in our editing process and we thank you once again for pointing this out. As you will see in the revised manuscript, we have added these references in the text as well. We have also added a supplementary reference within the introduction. We would like to mention that the introduction is mainly the result of the authors' personal contribution and was only in a very small percentage based on other sources. As you have also pointed out, there are not many studies at the moment regarding the impact of routinely dosing sdLDL subfractions on the overall cardiovascular risk. This is exactly why there are not as many references to previous research as we would have hoped for.
We hope we have answered all of your questions.
Best regards,
Reviewer 4 Report
Consider elaborating more in abstract regarding the LDL role in CVD with potential qualitative and quantitative analysis summary.
Summary of articles screened, number of patients in total, genetic (i.e, top 5 genes involved in regulating LDL) and demographic factor in this review etc. consider including it in abstract to set the tone.
Well developed flow of information on LDL, VLDL, Apo-B and it’s relation with CVD. Good discussion on future of this therapy. Consider writing more about gene specific therapies in Cardiovascular and identified genes.
Overall great article. No further major comments.
Author Response
Dear Reviewer,
Thank you very much for your kind and useful suggestions. We have added a paragraph regarding gene-specific lipid-lowering therapies. We would like to mention that we have only mentioned gene-related therapies and potential treatment targets to emphasize the importance of further studying the particularities of sdLDL particles and their overall impact in estimating cardiovascular risk within the general adult population. However, these areas of interest did not represent our main focus when we designed this review paper, mainly because there are currently only a few studies on this subject, most of them still ongoing at the moment, which is why we cannot go into much detail regarding this area of research. We are, however, studying this topic and eagerly awaiting the preliminary results from the ongoing clinical trials.
Best regards,
Round 2
Reviewer 1 Report
The authors have adressed all my comments and improved the quality of the paper
Author Response
Dear Reviewer,
Thank you once again for your kind review and your extremely useful recommendations. We are glad to have been able to address all your previous comments regarding our paper.
Best regards,
Reviewer 2 Report
I Inspect the corrected manuscript noting that suggestions and corrections have been accepted and applied. However, more corrections are needed, which are mentioned below, please review them and, and if so, apply them to the manuscript.
1) figure 1:
- Fix schema descriptors, some appear cut off
-In the image caption make a brief description of the figure mentioning the elements that make it up, it could be with a smaller letter size.
2) Line 94 Missing figure header
3) Line 112. rewrite “1.8….” It must be 1.8
4) Homogenize the writing of figures. For example, in line 123 it is write “…10 001..” while in line 130 it is write “…12,064..”
5) Renglón 159, escribir “…Yen et al. (2022) [12] en lugar de “…Yen et al….”
6) Line 190. Remove space in “…HDL -C…”
7) Referencias
arrange and homogenize the citations since they are in different formats, which also do not correspond to those indicated by the authors' guide. To write the quotes, take reference 11 as an example (check the authors' guide for the correct way to make a bibliographical citation).
Gurevitz, C.; Auriel, E.; Elis, A.; Kornowski, R. The Association between Low Levels of Low Density Lipoprotein Cholesterol and Intracerebral Hemorrhage: Cause for Concern? J. Clin. Med. 2022, 11, 536. https://doi.org/10.3390/jcm11030536.

Author Response
Dear Reviewer,
Thank you for your detailed and useful recommendations. As you will see in the revised manuscript, we have made all the recommended corrections and have edited the figure that you have mentioned. We have also updated the references section to respect the journal's guidelines. We hope to have answered all your concerns.
Best regards,